

# Long vs. Short: Understanding the dynamics of persistent summer hot spells in Europe

Duncan Pappert[1,2], Alexandre Tuel[3], Dim Coumou[4], Mathieu Vrac[5], and Olivia Martius[1,2,6]

[1]Institute of Geography, University of Bern, Bern, Switzerland
[2]Oeschger Centre for Climate Change Research (OCCR), University of Bern, Bern, Switzerland
[3]Galeio, Paris, France
[4]Institute for Environmental Studies, Vrije Universiteit Amsterdam, Amsterdam, the Netherlands
[5]Laboratoire des Sciences du Climat et de l'Environnement, CEA-CNRS-UVSQ, Université Paris-Saclay, Gif-sur-Yvette, France
[6]Mobiliar Lab for Natural Risks, University of Bern, Bern, Switzerland

**Correspondence:** Duncan Pappert (duncan.pappert@unibe.ch)

**Abstract.** The persistence of surface hot spells in Europe on subseasonal timescales can lead to significant socio-economic impacts. Here, we adopt a regional perspective to compare the dynamical features associated with long-lasting (persistent, 12 - 26 days) and short-lived (4 - 5 days) regional-scale hot spells over Europe during summer using the ERA5 reanalysis. We identify six coherent regions in Europe (Southwestern Europe, Western Europe, Central-Southern Europe, Northern Europe, Eastern Europe and Northwestern Russia) defined by the clustering of gridcells which experience long-lasting hot spells at the same time. Temperatures are averaged across these regions for an analysis of hot spells in SW and W Europe.

In SW Europe, persistent hot spells are tightly linked to antecedent soil dryness. Significant soil moisture anomalies are present in the weeks prior to and during the hot spells, but not prior to short hot spells. Persistent hot spells are associated with larger and higher magnitude positive blocking frequency anomalies compared to short spells as well as a significant positive frequency anomaly of cutoff lows upstream and south-west of the region, while the jet stream is shifted northwards. Large-scale anticyclonic Rossby wavebreaking over Europe and the Mediterranean is also often associated with persistent hot spells in SW Europe. During short spells the upstream jet is located further south and the upstream wavetrain is more zonally oriented, pointing to a more transient large-scale upstream flow configuration matching the more transient nature of the spells.

In W Europe persistent hot spells are marked by strong land-atmosphere coupling, leading to intense soil desiccation during the events but no significant soil moisture anomalies prior to the events. A lower wavenumber Rossby wavetrain compared to the short spells indicates a more stationary upper-level flow during persistent spells. High blocking frequency and recurrent Rossby wave packets (RRWPs) feature in 87% and 60% of persistent events in this region, respectively. During short spells the upstream jet over the Atlantic extends further east and the upstream cyclone frequency is significantly higher than in the climatology, pointing to the important role of cyclones for the termination of short hot spells.

In both regions several dynamical mechanisms (blocking, RRWPs, cutoff lows) are contributing to persistent spells; in 80% or more of the cases more than one type of mechanism was involved. The sequence of drivers during the persistent spells varies across spells. In both regions short spells are associated with a more transient flow situation upstream over the North Atlantic.





## 1 Introduction

During summer, anomalously high temperature conditions can sometimes persist for weeks. Europe has faced a number of
such events of prolonged extreme heat, with notable examples including the devastating heatwave of 2003 (Black et al., 2004;
Trigo et al., 2005; García-Herrera et al., 2010), the severe 2010 heatwave in Western Russia (Barriopedro et al., 2011; Di Capua
et al., 2021), and the Northern European heatwave of 2018 (Vogel et al., 2019; Drouard et al., 2019; Yiou et al., 2020). These
persistent extreme surface weather events resulted in enormous impacts by disrupting natural, economic and social systems
(De Bono et al., 2004; Zuo et al., 2015). Heatwaves are an established risk factor for human health and lead to additional deaths
(Robine et al., 2008; Muthers et al., 2017; Adélaïde et al., 2022; Ballester et al., 2023); they can increase the number of forest
fires (Porfiriev, 2014; Sutanto et al., 2020), damage infrastructure (Nguyen et al., 2010; Forzieri et al., 2018), and cause energy
shortages (Miller et al., 2008; Pechan and Eisenack, 2014), water scarcity (Belleza et al., 2023) and crop failure (Bastos et al.,
2014; Albergel et al., 2019). The longer their duration, the greater their impacts (Polt et al., 2023; Tuel and Martius, 2024),
emphasising the importance of further understanding their nature and improve their predictability.

There is substantial literature on extratropical summer heatwaves and their driving mechanisms, often focusing on their
frequency and intensity (see e.g. Perkins, 2015; Horton et al., 2016; Barriopedro et al., 2023). However, fewer studies have
isolated and explored the 'duration' dimension of these events. Heat extremes exhibit a wide range of possible durations. They
may last only a few days, persist beyond synoptic timescales for up to several weeks, or sometimes even endure for months.
They may occur as a single, continuous exceedance of an extreme threshold (quasi-stationarity) or be comprised of multiple
repeated occurrences separated by brief intervals (recurrence) (Tuel and Martius, 2023). In this study, we use the term 'hot
spell' to encompass all these instances of exceptionally warm temperatures.

Knowledge regarding longer-lasting hot spells and their associated drivers and predictors remains limited. This is particularly
the case for those hot spells occurring within the subseasonal timescale, which shall henceforth be referred to as 'persistent'
(or 'long') hot spells. That said, a number of studies have proposed processes and mechanisms that can contribute to the higher
persistence of mid-latitude hot spells.

Perhaps the most widely recognised driver of persistent surface heat is land-atmosphere feedbacks (e.g., Fischer et al., 2007;
Felsche et al., 2023). As a slowly varying variable, soil moisture serves as a crucial water and energy storage that couples
land and atmosphere. This characteristic enables soil moisture to modulate near-surface weather over timescales of weeks
to months, much beyond that life-time of individual synoptic systems (Koster and Suarez, 2001; Wu and Dickinson, 2004;
van den Hurk et al., 2012). Precipitation deficits over land can lead to dry soil anomalies, and as a drought develops, the
reduction in evapotranspiration leads to lower moisture availability, allowing more of the sun's energy to be used for sensible
heating, inducing an increase in near-surface air temperatures (Seneviratne et al., 2010). The positive temperature anomalies
obtained through the land surface feedback also exert a dynamical feedback by warming the boundary layer and increasing
the geopotential height anomalies. Under favourable conditions, e.g. once an anticyclone establishes itself in large-scale flow,
this land-atmosphere interaction ensures that the atmospheric forcing requires less 'effort' to reach extreme temperatures. The



resulting above-average number of hot days could mean a higher chance of obtaining hot spells, perhaps even persistent ones (Lorenz et al., 2010; Müller and Seneviratne, 2012).

Though soils can act as temperature amplifiers/dampeners, the occurrence of hot spells is primarily influenced by the more rapidly-varying atmospheric flow. Indeed, persistent hot spells have often been linked to the occurrence of atmospheric blocking (Black et al., 2004; Drouard and Woollings, 2018) and associated double jet flow structures (Perkins, 2015; Rousi et al., 2022). By definition long-lasting and quasi-stationary systems that disrupt the westerly zonal flow, blocks have a well-established link to prolonged surface weather (Röthlisberger and Martius, 2019; Kautz et al., 2022), among them quasi-stationary amplified circumglobal waves (Kornhuber et al., 2017). Weather recurrence, too, can contribute to persistence (see Tuel and Martius, 2023, for a review). For example, synoptic-scale recurrent Rossby wave packets, i.e., troughs and ridges amplifying repeatedly at the same longitudes (Röthlisberger et al., 2019; Ali et al., 2021, 2022). Highly nonlinear processes such as Rossby wave breaking tie into several of the above-mentioned structures and can also be relevant for prolonged surface weather, though they have more often been studied in connection with persistent precipitation and flooding (Grams et al., 2017; Steinfeld and Pfahl, 2019; Mohr et al., 2020; Tuel et al., 2022; Thompson et al., 2024). There is also evidence to suggest that weaker storm track activity is associated with persistent surface heat, particularly over western Eurasia (Pfleiderer and Coumou, 2018).

Extreme events falling within the subseasonal timescale are influenced by atmospheric flow configurations, which may extend beyond the usual lifetime of synoptic weather systems, and the slower-evolving climate variables such as soil moisture (Vitart and Robertson, 2018). Persistent hot spells, therefore, also hold significant interest for the subseasonal to seasonal prediction (S2S) scientific community due to their potential for providing windows of high predictability for extended-range forecasts (White et al., 2017; Mariotti et al., 2020). Nevertheless, that which gives persistent hot spells predictive potential is the same that renders them notoriously challenging to forecast skillfully — namely, the complex interaction of multiple physical links at different temporal scales.

While still valuable, the focus on either individual mechanisms or single case studies can lead to a fragmented understanding of the clearly multifaceted dynamics that create persistent hot spells. Moreover, we expect the relevance and relative contribution of this 'ensemble of processes' to vary depending on the affected region (e.g., Pfleiderer and Coumou, 2018; Tuel and Martius, 2024) and there remain gaps in our knowledge in this regard. Understanding the spatio-temporal characteristics and the processes that contribute to the persistence of hot spells in Europe is vital for their accurate representation in models, thereby better estimating their impacts and enhancing forecasting capabilities (Jacques-Dumas et al., 2022; Domeisen et al., 2023).

To address the research gaps, we conduct a systematic analysis of the dynamics linked to persistent hot spells in selected regions of Europe based on a physically-consistent regionalisation. We want to understand why some surface extreme temperature events last longer than others, as well as gauge the extent of what the modern observational record can tell us. For this we compare and contrast long-lasting spells to short-lived ones to better understand the dynamics determining differences in duration. Much in the same way that Drouard and Woollings (2018) did in their study on atmospheric blocks, our purpose is to characterise persistent hot spells through comparison with short ones and to identify significant common ingredients among



both types, which might improve the prediction of the former. To our knowledge, long- and short-duration heat extremes have so far not been examined in such a framework in the literature.

Section 2 introduces the data and the methodological approach underpinning the study. Results from the regionalisation and hot spell identification are shown in Section 3.1. Sections 3.2 and 3.3 provide a comprehensive analysis of a number of hot spell drivers for two selected regions and interpret the results. We begin by examining the link between dry soils and hot spells. In a second part, we use composite analysis to investigate atmospheric variables and processes associated with near-surface temperature persistence. Finally, in Section 3.4, we illustrate the complexity and variability of persistent hot spells through some examples. Findings and final perspectives are summarised in Section 4, along with avenues for further research.

## 2    Data and Methods

This study uses data from the ERA5 reanalysis over the Northern Hemisphere for 1959-2022 (Hersbach et al., 2020) at 0.5°x0.5° spatial resolution and a daily temporal resolution. Daily means are computed from 6-hourly data. The data encompass 2m temperature, 500hPa geopotential height (Z500), wind speed at 300hPa (W300), total precipitation (TP), and top-1m soil moisture (SM). SM is determined using a depth-weighted average of the reanalysis' volumetric soil water content layers 1-4, expressed in water percentage by volume. We use the contour tracking tool ConTrack developed by Steinfeld (2020) and based on the work by Schwierz et al. (2004) to detect blocking and cyclone activity. Blocks are identified as regions of 500-150 hPa vertically integrated potential vorticity (PV) anomalies below the climatological 10th percentile, with a 70% contour overlap between 6-hourly timesteps for a minimum of 5 days. Cyclones are tracked as closed contours of 6-hourly mean sea level pressure (MSLP) anomalies below the climatological 10th percentile; the MSLP field has been previously filtered for frequencies in the 2.5- to 6-day range using a Butterworth bandpass filter. PV cutoff vortices, as defined by Wernli and Sprenger (2007), are detected using the tracking tool developed by Kaderli (2023). A cutoff cyclone is identified following the 2 PVU contour line on five isentropic levels between 350-330K and then vertically averaged, subsequently retaining only gridcells where the cutoff structure is found on at least two levels. Finally, to assess the presence and strength of recurrent Rossby wave packets (RRWPs), we compute the R metric as introduced in Röthlisberger et al. (2019).

Standardised anomalies are calculated for a given day with respect to a mean and standard deviation estimated from a 30-day 8-year moving window, similarly to Pfleiderer et al. (2019) and Tuel and Martius (2024). This procedure removes seasonality and long-term trends in the data.

### 2.1    Extreme temperature regionalisation

To reduce the dimension of the problem, we cluster the gridcells of our European domain (32-72°N,25°W-53°E) into distinct regions that experience long-lasting heat extremes simultaneously and thus would share a similar association with large-scale circulation. Before applying the clustering, we first apply a land-sea mask to the T2M fields to retain only gridcells over land, as we are interested in the impacts. We pre-process the daily data by calculating for each gridcell 3-week non-overlapping T2M



averages from the standardised deseasonalised and detrended anomalies. The time-averaged series are then binarised, with the 3-week periods above each gridcell's climatological JJA 95th percentile being set to 1 and those below to 0. We cluster gridcells using the pairwise Jaccard distance matrix, which measures the total number of dissimilar entities between sets divided by the total number of entities:

$$D_J(\mathbf{x}, \mathbf{y}) = 1 - J(\mathbf{x}, \mathbf{y}) = \frac{|\mathbf{x} \cup \mathbf{y}| - |\mathbf{x} \cap \mathbf{y}|}{|\mathbf{x} \cup \mathbf{y}|} \tag{1}$$

where $J(\mathbf{x}, \mathbf{y})$ is the so-called Jaccard coefficient (see Jaccard, 1912; Choi et al., 2010). This distance metric is ideal for our purposes, as we are interested in the few 1s — meaning the extreme events — and not the many 0s (non-events). Also, the Jaccard distance corresponds to the level of event synchronicity between two binary time series, meaning that the number of clusters can later be defined by requiring that the average synchronicity for each cluster should be above some minimum value. For the clustering procedure itself, we compute a simple agglomerative hierarchical clustering with the average linkage method, the implementation of which is provided by the `SciPy` Python package and uses the unweighted pair group method with arithmetic mean (UPGMA) algorithm (Virtanen et al., 2020). In this approach, there is no need to pre-specify a number of clusters; instead, the resulting dendrogram can be truncated at a desired distance that corresponds to a degree of event co-occurrence. Accordingly, at the truncation distance t=0.8, we identify clusters corresponding to regions that are completely off continental Europe (excluding the British Isles). We then remove the corresponding gridcells and re-run the clustering procedure. Truncating the dendrogram at t=0.875, indicating a 12.5% level of event synchronicity, we obtain 6 distinct European clusters (see Fig. 2a).

For the present study, we select two regions intended to be illustrative of our methodology and the central messages we aim to communicate. Thus, the dynamical analysis in Sections 3.2, 3.3, and 3.4 is centred on clusters 1 and 2, representing Southwestern Europe and Western Europe, respectively. Figures pertaining to the analysis of other regions can be found in Section 4 of the Supplementary Materials.

## 2.2 Definition of long- and short-duration hot spells

The result is such that each region now has its own distribution of hot spell durations, ranging from the more common short-duration events and the comparatively fewer (rarer) long-duration events. We define long (persistent) spells as events lasting between 12 and 26 days and short spells as lasting between 4 and 5 days (see Fig. 2b). We disregard very short hot spells (1-2 days) as we want our short events to align more closely with the average lifetime of synoptic weather systems. The choice to adopt a broad 14-day range for defining persistent (long) hot spells is somewhat subjective, as there is no agreed-upon definition. However, this decision is intentional, aimed at encompassing the subseasonal timescale as well as increasing the sample size of events. Similarly, opting for a threshold of 1 standard deviation rather than a more extreme one is also due in part to bolster the number of cases — nevertheless, these events can still be considered periods of moderately high temperatures. With the long and short sets of events defined, we are now in a position to make comparisons and explore reasons for such differences in duration.



Our hot spell definition does not differentiate between quasi-stationary and recurrent behaviour in near-surface weather.
As such, it does not distinguish between, for example, an intense heatwave that surpasses the extreme threshold consistently
throughout its lifetime and an event comprised of multiple waves occurring in rapid succession. From an impacts perspective,
it is reasonable to consider both flavours of persistence together, since both can cause prolonged and impactful surface weather
conditions (Tuel and Martius, 2023).

## 2.3   Composites and comparisons

Through composite analysis, we discern the characteristics shared by persistent hot spells and compare them with those of short
spells. The meteorological fields are composited by initially computing the mean of the days corresponding to the individual
spells and subsequently by computing the mean of the spell means in the sample set. Nevertheless, meaningful comparisons
are made difficult by two factors: firstly, the scarcity of the (rarer) detected long spells, constituting only 28% and 41% of the
number of short spells in regions 1 and 2; secondly, compositing events that last as long as 2-3 weeks captures more variability
than events averaged over just 4-5 days. To address this issue, we randomly subsample 4-5 day periods in the long spells as
many times as there are short spells. The subsampling is performed 100 times and the medoid of the resulting 100 composites
— i.e. the most representative one in the set — is retained for the comparison with the short spell composite. This approach
ensures a more balanced comparison between events of different duration and unequal sample size. Thus, the compositing
operation for short hot spells can be formulated in the following way:

$$comp_{short} = \frac{1}{N} \sum_{j=1}^{N} \left( \frac{1}{d} \sum_{i=1}^{d} s_{ij} \right) \tag{2}$$

and for the long hot spells:

$$comp_{long^*}^{(k)} = \frac{1}{N} \sum_{j=1}^{N} \left( \frac{1}{d} \sum_{i=1}^{d} l_{ij}^* \right), \quad \text{for } k = 1 \text{ to } 100 \tag{3}$$

$$comp_{long^*,medoid} = \arg \min_{y \in \mathbf{comp_{long^*}^{100}}} \sum_{k=1}^{100} D\left( y, comp_{long^*}^{(k)} \right) \tag{4}$$

where $d$ and $N$ are the number of days in the spell and the number of events; $D$ is the distance function in Eucledian space; $s_{ij}$
and $l_{ij}^*$ represent a given day $i$ in event $j$ of the short and long ('*' = subsampled) hot spells, respectively. Figure 1 provides
a schematic illustration of this methodology. It is important to note that the long-spell subsampling pertains mainly to the
analysis of atmospheric variables during the extreme events themselves (Sect. 3.3). The procedure is not applicable when
aligning events around their beginning date, as in the case for the SM analysis (Sect. 3.2). When examining the probability of
hot spells associated with dry soils, we bootstrap the short spell sample to match the (scarcer) climatological frequency of the
long spells. Once more, this approach allows for meaningful comparisons between the two sets of events for each region.





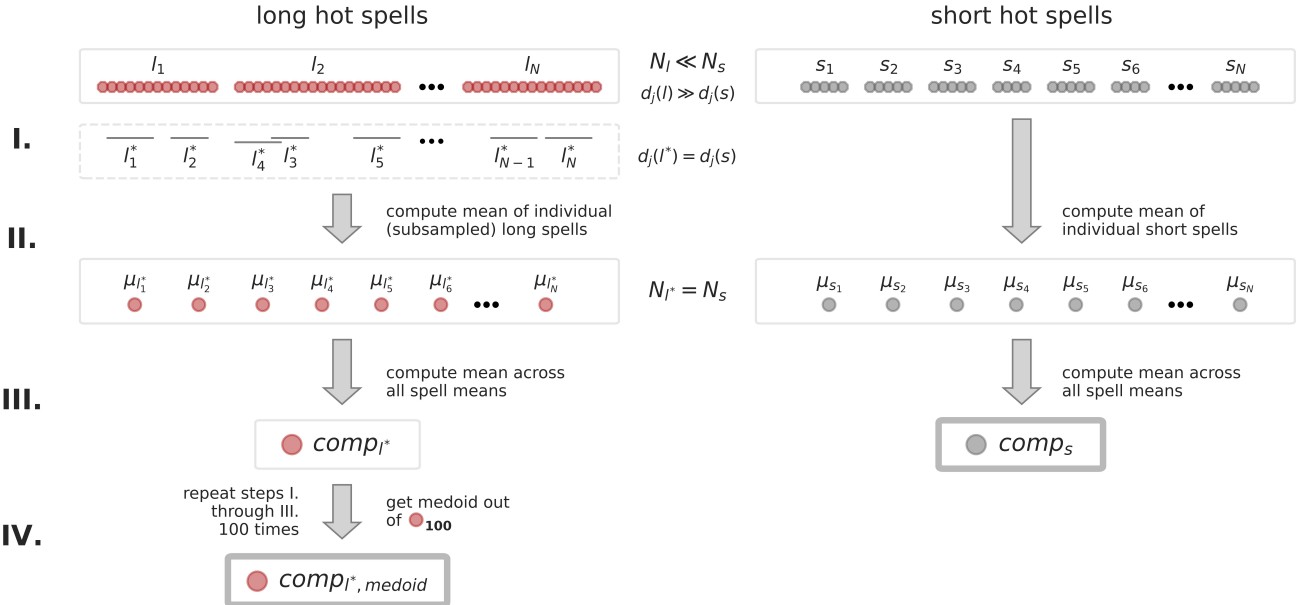

**Figure 1.** Flowchart of methodology for the composite comparison of long and short hot spells used in Sect. 3.3. Step I is the random subsampling of long-duration events; step II is the computation of spell means; step III is the computation of the mean of means; step IV is the selection of the medoid out of the 100 composites from different subsampled sets and pertains only to the long spells. $N$ represents the spell sample size and $d_j$ is the duration of individual events.

We test the statistical significance of the composite anomaly fields with a Monte-Carlo approach, by assessing at each grid point their rank among 1000 anomalies obtained from randomly generated events. For the long spell fields, we consider grid cells significant if they are identified as such in at least two-thirds of the 100 subsampled composites. The random dates for the Monte Carlo sampling are drawn from the summer distributions of occurrences for both the long and short spells in each region. Resulting p-values are subsequently adjusted to account for the false discovery rate (FDR) arising from multiple hypothesis testing (Wilks, 2016).

## 3 Results & Discussion

### 3.1 Regions and hot spells

The six clusters that result from the extreme temperature regionalisation (Fig. 2a) can be broadly defined as: 1) Southwestern Europe, 2) Western Europe (incl. British Isles), 3) Central-Southern Europe, 4) Northern Europe / Scandinavia, 5) Eastern Europe / W Russia , and 6) Arctic / NW Russia. These clusters align closely with areas frequently studied in past and current heatwave research (Stefanon et al., 2012; Zschenderlein et al., 2019; Schielicke and Pfahl, 2022; Felsche et al., 2023; Pyrina and Domeisen, 2023). Our regionalisation also demonstrates some robustness, as regions exhibit minimal qualitative differences





when the clustering to six final clusters is conducted using 2-week T2M averages and/or a less extreme threshold. We refer the reader to Section 1 of the Supplementary Materials for a more detailed exploration of the robustness of the regionalisation.

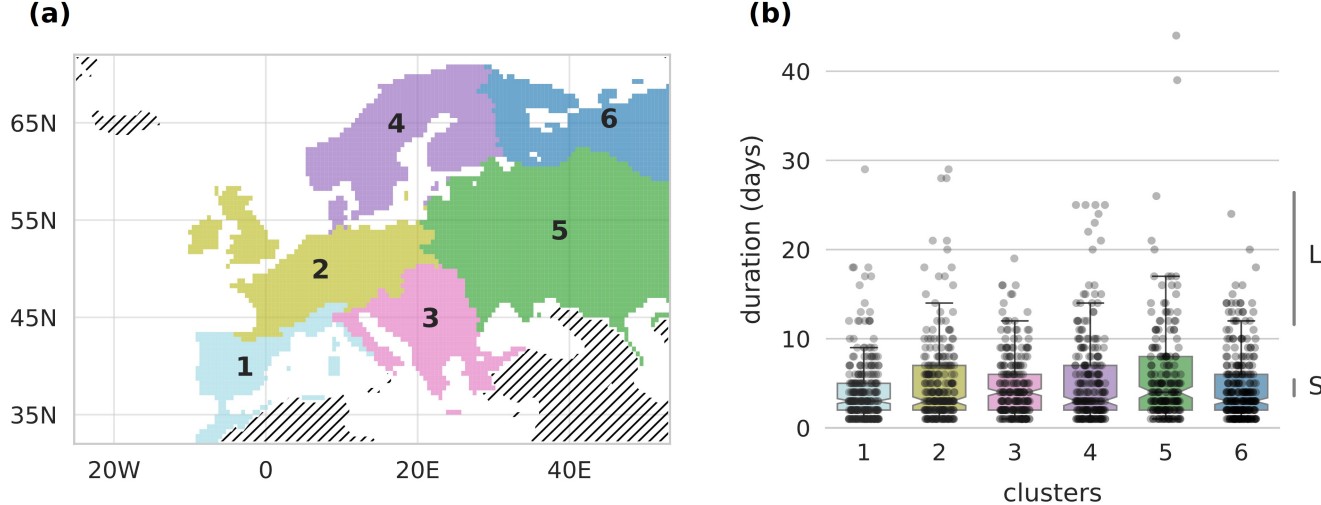

**Figure 2.** (a) Map of the six European regions obtained with the cluster analysis. (b) Distribution of hot spell durations across clustered regions. Each dot represents a detected event, with 'L' denoting the range that includes long spells (12 to 26 days) and 'S' marking the short spells (4 to 5 days).

Figure 2b depicts the distribution of hot spell durations across each region; relevant statistics regarding the boxplots are provided in Table 1. Short spells are more numerous and therefore also visibly closer to the median, whereas long spells are in the tail of the hot spell distributions, indicating their relative scarcity. The mean spell duration is around 4-5 days, which would actually qualify such events as having an average duration. Nevertheless, we consider these to be short hot spells relative to the subseasonal event lifetime of interest.

Both regions in S Europe have the highest number of short spells, and the fewest long spells (this last point including W Europe). In contrast, N/E Europe, specifically regions 4, 5, and 6, average nearly twice as many long spells as those in W/S Europe (regions 1, 2, and 3).

The varying hot spell statistics of each cluster emphasise the importance of performing the extreme temperature regionalisation. Understandably, however, the actual number of detected hot spells is somewhat susceptible to the exact hot spell definition being used, nevertheless, the broad characteristics remain consistent. We refer the reader to the Supplementary Materials, Section 2, for an exploration of how different approaches to identify a region's hot spell affects the selected events. The analysis in the following sections covering SW Europe (region 1 in Figure 2) and W Europe (region 2 in Figure 2).





**Table 1.** Summary statistics of the identified hot spells for each cluster from the regionalisation.

| clusters | median spell duration | mean spell duration | number of short spells (4-5 days) | number of long spells (12-26 days) | total number of spells | long / short | long / total |
|---|---|---|---|---|---|---|---|
| *1* | 3 | 4.2 | 57 | 16 | 280 | .28 | .06 |
| *2* | 3 | 5 | 39 | 15 | 258 | .38 | .06 |
| *3* | 4 | 4.5 | 52 | 13 | 255 | .25 | .05 |
| *4* | 3 | 5.3 | 33 | 26 | 242 | .79 | .11 |
| *5* | 4 | 5.7 | 41 | 24 | 210 | .59 | .11 |
| *6* | 3 | 4.6 | 44 | 24 | 272 | .55 | .09 |

## 3.2 The link to soil moisture

Both within and across regions 1 and 2, clear differences can be observed between long and short hot spells with regard to the temporal evolution of SM anomalies in the run up to the events (Fig. 3).

In SW Europe (region 1), long spells are frequently preceded by dry soils, as evidenced by the fact that most values in the boxplots are negative (Fig. 3a left panel). The average extent of dryness across long spells is statistically significant as far back as eight to six weeks prior to the events. Furthermore, the overall spread of anomalies is smaller for long spells compared to short ones. While long spells are almost exclusively associated with very dry soil anomalies in the preceding weeks, short spells are preceded by a wide variety of soil conditions, from rather wet to very dry (Figure 3a).

In W Europe (region 2), the spread of anomalies is large for both long-lasting and short-lived hot spells (Fig. 3a right panel). Both types of events can occur after periods of anomalously high and low SM, with an overall tendency towards drier soils in the last five weeks for long spells. Indeed, the mean evolution of SM anomalies falls towards increasingly negative values up until the onset of long spells, though average values are not significantly different from zero.

Once the hot spells in both regions commence — i.e., the period represented in Figure 3a by the thinner boxes on the right of each boxplot group — there is a marked jump to more negative values due to the positive soil-temperature feedback taking effect. The sudden strong desiccation is more pronounced in W Europe than in SW Europe. In both areas, this drop marks a shift to statistically significant average negative SM anomalies, except for short spells in W Europe. The onset of long spells in this region is highly unlikely to experience wetter-than-average SM.

Figure 3b illustrates the likelihood of region-averaged SM anomalies falling below specific percentile thresholds (SM < x) during different periods leading up to and at the onset of hot spells. In both regions and across all considered periods, the proportion of long spells associated with soil dryness (even extreme dryness) is almost always either above the upper quartile or completely exceeding the 1.5 interquartile range (IQR) value of the probability distribution for short spells – under the assumption of equal climatological frequency with long spells (see boxplots in Fig. 3b left panel). In other words, only in rare cases are short hot spells linked to dry soils to the same degree that persistent events are.







**Figure 3.** Top row (a): Region-averaged standardised soil moisture anomalies in the weeks leading up to the hot spells in regions 1 (left panel) and 2 (right panel) for long- (coloured) versus short-duration (faded colour) hot spells. The boxplots represent 2-week overlapping mean SM anomalies prior to the beginning of all spells (8/6, 7/5, 6/4, 5/3, 4/2, 3/1, 2/0 weeks prior), with the final thinner boxplot denoting the values during the first 4 days of the spell. The circles connected by the line are the means of the individual distributions and the filled ones indicate mean anomaly values that are statistically significant at a confidence level of 95%. The dashed horizontal lines stand for the means across each boxplot grouping. Bottom row (b): Probability of long and short hot spells in SW Europe (R1) and W Europe (R2) being preceeded by 60-day and 20-day periods of dry soils as well as during the first 4 days of the events themselves. The thresholds 'x' for soil dryness are the mean $\mu$, $30^{th}$, $20^{th}$, and $10^{th}$ percentiles of the deseasonalised and detrended AMJJA SM climatology 1959-2022.





Generally, hot spells in SW Europe are more likely to have been preceded by dry soils than those in W Europe, except for
when antecedent SM averages fall below the respective climatological 10% quantile relative to the climatology of each region.
The desiccating effect of the positive soil-temperature feedback at the onset of hot spells (first 4 days) is evident in both regions,
marked by a shift to higher probabilities. As also shown in Figure 3a, the relative extent of this pre- to post-spell shift is greater
for W Europe (Fig. 3b). Overall, persistent hot spells in both regions exhibit a clear — albeit differing in nature — link to SM
deficits. For SW Europe the 2-month antecedent dryness in long spells marks the difference to short spells, while further north
in W Europe, the notable factor is the contrasting spread of anomalies at spell onset.

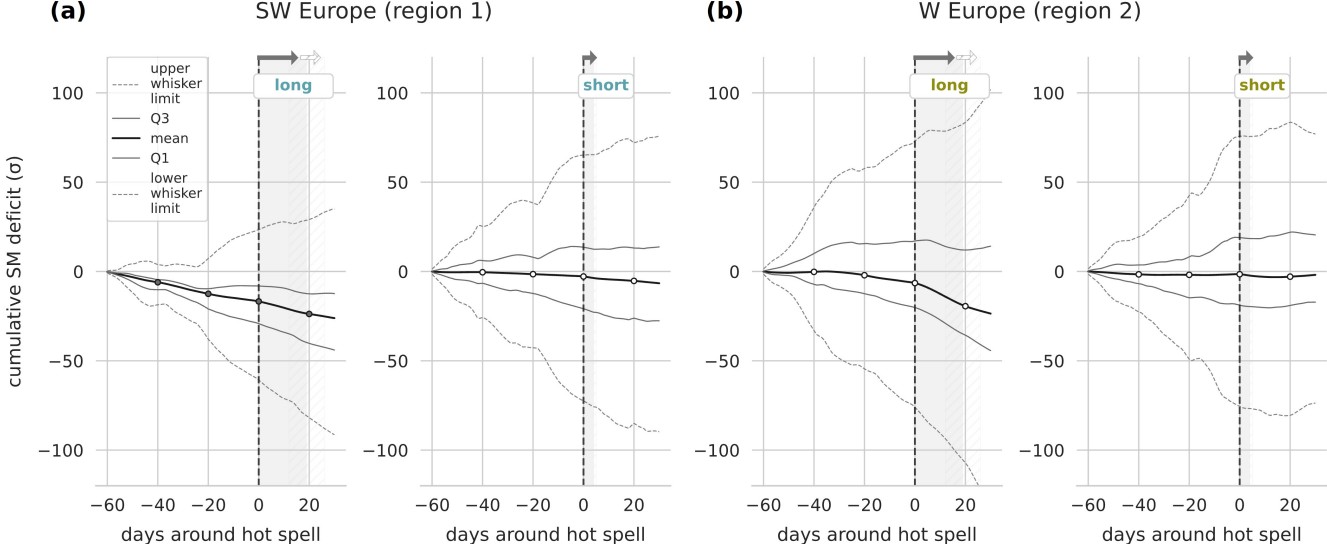

**Figure 4.** Cumulative SM curves calculated with standardised anomalies of volumetric soil water content around long and short spells in (a)
SW and (b) W Europe. The x axis ranges from 60 days before to 30 days after hot spell onset. The curves denote the mean (thick black), the
$1^{st}$ and $2^{nd}$ quartiles (gray), and the lower and upper whisker limit values (thin gray dashed). Filled circles indicate timesteps at which the
mean value is significantly different from zero at the 95% level.

Complementing this picture are the results for the accumulated SM deficit shown in Figure 4. Long spells in SW Europe
show a significant constant depletion of SM from 60 days prior to beyond the events' decay, with the bulk of values showing
this deficit as well as statistically significant mean values. Conversely, there is no significant deficit in the lead up to long spells
in W Europe. However, during these events, a pronounced accumulated SM deficit develops as the hot spell progresses, though
the values themselves are not significantly extreme. In both regions, short spells show no preferred tendency in the antecedent
SM, with progressive depletion and accumulation in the months prior being equally likely precursors.

As explained in the introduction, there is strong evidence for antecedent dry soils to extend the length of heat extremes in
Europe (Brabson et al., 2005; Lorenz et al., 2010). For instance, through modelling experiments, Lorenz et al. (2010) found that
antecedent dry conditions in the Iberian Peninsula increase both the length and amplitude of extreme threshold exceedances.



A semi-arid region such as SW Europe has a relatively high SM memory (Seneviratne et al., 2010). An unusually long and intense drought in this region, therefore, would cause a cumulative depletion of water in the soils such that even more time is needed for the drought to recover and for the 'bucket' to fill to levels where the energy flux partitioning is not favouring sensible heating (Fig. 4a). Until then, the positive warming feedback may continue unless the atmospheric circulation ends the spell by, for instance, causing above-normal precipitation or cold air advection into the region. This is one hypothesis that

could explain why long spells in SW Europe are preceded by significant soil dryness up to two months prior. Some short hot spells are preceded by substantial SM deficits (Fig. 4a) and have the potential to last longer, but in these cases the atmospheric forcing did not allow the hot spell to persist.

Located further north in a wetter regime, W Europe's hot spells do not have a statistically significant link to antecedent SM deficits (Fig. 4b). Hot spells in W Europe are just as likely to follow wet soil conditions Rouges et al. (2023), making low

SM less of a common precondition for long spells as for SW Europe. There is a pronounced shift to significantly negative SM anomalies at long spell onset (Figs. 3-4), especially in W Europe. The difference in the association with dry soils between long and short spells is substantial. While a very strong positive land-atmosphere feedback and the resulting extremely high temperature could eventually terminate a heatwave due to convection onset (Zhang and Boos, 2023), this effect may not come into play for our long hot spells, as the temperatures needed for convective instability are much higher.

While low SM favours extreme temperature persistence, our analysis shows that, although a common precondition, it is not a necessary prerequisite for long spells, and even less so for short ones. Figure 3 showed that half of the time in both regions, short hot spells were associated with negative SM anomalies yet the events remained short-lived. The additional Figure S10 (Supplementary Materials, Sect. 3) indicates that there is no monotonic relationship between SM anomalies and hot spell duration; in fact, SM anomalies around the event better explain its mean intensity rather than its duration. This suggests

that something other than the land-atmosphere feedback must be determining the duration of hot spells. Indeed, researchers acknowledge that heatwaves are mainly atmospheric-driven events (Perkins, 2015; Horowitz et al., 2022), with their occurrence and duration strongly determined by the large-scale circulation patterns (Lorenz et al., 2010).

### 3.3   Atmospheric drivers

Figures 5 and 6 display spatial composites for several atmospheric variables and features during the long and short hot spells

for SW Europe (region 1) and W Europe (region 2).

#### 3.3.1   Upper-level Rossby wavetrain orientation and wavenumber

More substantial than the difference in the magnitude of the upper-level circulation anomalies is the difference in their spatial extent and orientation. This is most apparent in the Z500 anomaly composites for W Europe (Fig. 6a,b), where the large-scale ridges and troughs that form the Rossby wavetrain pattern have a greater longitudinal extension during long spells than short

ones. The anomalies are mostly not significant, suggesting that sample variability likely affects the location of the trough and ridge centers. Nonetheless, the footprint of the longer wavelengths is evident and indicative of a more stationary flow, leading





**Figure 5.** Spatial composites for long (left column) and short (right column) hot spells in SW Europe (region 1). The variables Z500, W300, and TP represent geopotential height at 500 hPa, wind speed at 300 hPa, and total precipitation, respectively. Filled contours depict the anomaly field. The contours in e,f (green) show the corresponding absolute wind field to the anomalies (24 and 28 m/s). The other contours (black) show the mean MJJAS frequency, specifically: 10, 12, and 14% (c,d); 4, 7.5, and 10% (g,h); 14, 17, and 20% (k,l). Dashed (dotted) hatching in the long spells denotes areas where the anomalies are significant at a 95% confidence level in over one-third (two-thirds) of the 100 subsampled composites; dotted hatching in the short spell composites denotes significant anomalies at the 95% level.





to more persistent weather. Note that the 'whole-event' composites of the long spells show the statistical significance of the wavetrain anomalies (Fig. S20, Supplementary Materials, Sect. 5).

The Z500 wavetrains during hot spells in SW Europe also display this difference in wavelength (Fig. 5a,b). However, more relevant is the difference in orientation of the upper-level Rossby wave pattern. For the long spells there is has a marked NW-SE tilt typical for a weak waveguide and anticyclonic (Branstator, 2002). Rossby wave breaking (RWB) (Zhang and Wang, 2018; Zavadoff and Kirtman, 2019). While anticyclonic RWB is less persistent than cyclonic Rossby wave breaking (Thorncroft et al., 1993), the almost planetary scale of the anticyclonic wave breaking during long spells makes the structure more robust against cross-contour PV advection and barotropic instability and hence longer-lasting. Anticyclonic RWB results in a poleward displacement of the upper-tropospheric jet (Martius et al., 2007; Barnes and Hartmann, 2012) and hence a northward shift of the storm track and cyclonic activity would be deviated into N Europe, far away from the Iberian Peninsula. The upper-level wind field in the long spell composites shows a poleward shifted jet over the North Atlantic (Fig. 5e), although the positive anomalies themselves are not significantly different from zero. This picture contrasts with that of the short spells, which instead shows a small area of significantly strong winds located further southward, below the upstream trough ahead of SW Europe (Fig. 5f). Cyclone frequency during long spells, too, is slightly anomalously positive over the N Atlantic – close to where the storm track is climatologically most prevalent – though not significant (Fig. 5i).

### 3.3.2 Blocking

Atmospheric blocks are themselves synonymous with persistence (Rex, 1950; Woollings et al., 2018), even described as analogous to traffic jams in the predominantly eastward flow of the mid-latitudes (Nakamura and Huang, 2018). Their stationarity makes for a clear dynamical link to persistent surface temperature anomalies. Indeed, Röthlisberger and Martius (2019) found that the odds of a hot spell to survive into next day are increased by as much as 200 to 300% in SW Europe when co-occurring with a block. Such a high odds ratio is owed to blocks being climatologically rarer at these lower latitudes (Zschenderlein et al., 2019), meaning that the few times they do occur, their impact on hot spell persistence is substantial. This aligns well with the difference in blocking frequency and size in our composites (Fig. 5c,d). Blocks in W Europe, conversely, are significantly more persistent during compound hot and dry spells (Röthlisberger and Martius, 2019); this aspect, too, is consistent with findings from our SM analysis (Fig 4b, left panel).

The local positive blocking anomalies for long spells in W Europe are more meridionally extensive, contrasting with the short spells, for which the smaller positive anomaly is clearly bounded by negative frequency anomalies (Figs. 5-6c,d). Both in the blocking and wind composites, the subsampled long spells do not show statistical significance over the whole area, which could be indicative of the large variability in size, shape, and strength of the feature. That said, compositing over the whole duration of the persistent events leaves no doubt as to the relevance of large-scale blocking across their whole lifetime (Fig. S21, Supplementary Materials, Sect. 5).

Note that results are sensitive to the blocking definition and tracking method. For instance, when using blocks defined with Z500 instead of VIPV, the blocking frequency anomalies associated with hot spells remain within the same range, but the difference between long and short becomes by far more pronounced (see Fig. S11, Supplementary Materials, Sect. 3), which



would support our arguments even more. Part of the reason for the differing result could be that Z500 is a variable more tightly correlated with surface heat than VIPV (Chan et al., 2019), meaning the tracking algorithm potentially captured also subtropical anticyclones. The nature of the link to surface hot spells depends on the blocking index being used (Villiger, 2017).

### 3.3.3 Jet and storm track

Accompanying the blocks in W Europe is a splitting of the jet into two branches around the blocking anticyclone (Fig. 6e,f). The block size determines the extent of the northerly deflection of the jet streak.

The short spells in both SW and W Europe show a small area of significant positive W300 anomalies upstream of the hot spell regions and equatorward of the troughs over the western Atlantic (Figs. 5-6f). At the eastern edge of the positive wind anomalies we find statistically significant positive precipitation anomalies (Figs. 5-6h) and cyclone frequencies (Fig. 5-6j). The
cyclones and precipitation are signs that the hot spell is soon to end; they can be interpreted as evidence of the transience of the circulation and that variable weather will shortly propagate eastward into the hot spell region and lead to the decay of the hot spell.

### 3.3.4 Cutoff lows

The bottom-most panels in the composite figures show frequency anomalies of cutoff lows (Figs. 5-6k,l). These high-PV
vortices are detached from and equatorward of the main westerly current in the jet stream and thus potentially quite stationary. In the region over the Atlantic where cutoffs lows are most likely to occur, both spell types in SW Europe display a positive-negative-positive tripole pattern. Interestingly, the statistical signal is contrasting: the more remote dipole is significant for short spells but not for long spells. Conversely, the long spell composite shows a strong positive signal upstream of the hot spell region (as much as 10% more frequent than the MJJAS climatology). The cutoffs' position with respect to the hot spell
area suggests that the cutoff would advect warm air from lower latitudes into the hot spell region. This warm air can be moist or dry, depending on the exact position of the cutoff with respect to the Atlantic or the Sahara, i.e., oceanic or continental (Wernli and Sprenger, 2007). The cutoff could either be short-lived and arrive at a fortuitous moment to extend the lifetime of an ongoing hot spell or establish itself for a longer time as the upstream part of a quasi-stationarity omega block.

Thus, while upstream cyclones will eventually move in to terminate hot spells, the cyclonic activity of cutoff lows can
prolong them. It could be the latitudinal difference of these systems that explains these opposite effects.

High-PV cutoffs have only recently been studied in relation to heatwaves, though mostly focusing over W Europe and not in connection with event persistence (Noyelle et al., 2024). According to our composites, cutoffs do not appear to be relevant mechanism during long hot spells in W Europe.





**Figure 6.** Spatial composites for long (left column) and short (right column) hot spells in W Europe (region 2). The variables Z500, W300, and TP represent geopotential height at 500 hPa, wind speed at 300 hPa, and total precipitation, respectively. Filled contours depict the anomaly field. The contours in e,f (green) show the corresponding absolute wind field to the anomalies (24 and 28 m/s). The other contours (black) show the mean MJJAS frequency, specifically: 10, 12, and 14% (c,d); 4, 7.5, and 10% (g,h); 14, 17, and 20% (k,l). Dashed (dotted) hatching in the long spells denotes areas where the anomalies are significant at a 95% confidence level in over one-third (two-thirds) of the 100 subsampled composites; dotted hatching in the short spell composites denotes significant anomalies at the 95% level.




### 3.3.5 Summary

Figure 7 summarises region-specific information – some more robust than others – that amounts to a characterisation of persistent hot spells in a schematic for each region and spell type. An important feature of long spells in SW Europe are the extremely dry antecedent soils in the weeks prior, planetary-scale anticyclonic RWB, frequent blocking, and the above-average presence of cutoff lows. Long spells in W Europe stand out for the strong soil drying during the event, the longer wavelength of the Z500 wavetrain, meridionally extended blocks, and the absence of anomalous cyclonic activity upstream.

Short spells display an overall more transient situation upstream over the North Atlantic as seen in the wind speed, cyclones, shorter wavelengths and fewer blocks. These individual strands of evidence hint that the atmospheric flow is set up such that the upstream disturbance is moving east into the hot spell region – a situation prevented in the case of long spells. This also demonstrates that the upstream dynamics over the North Atlantic are very relevant for understanding hot spell persistence.

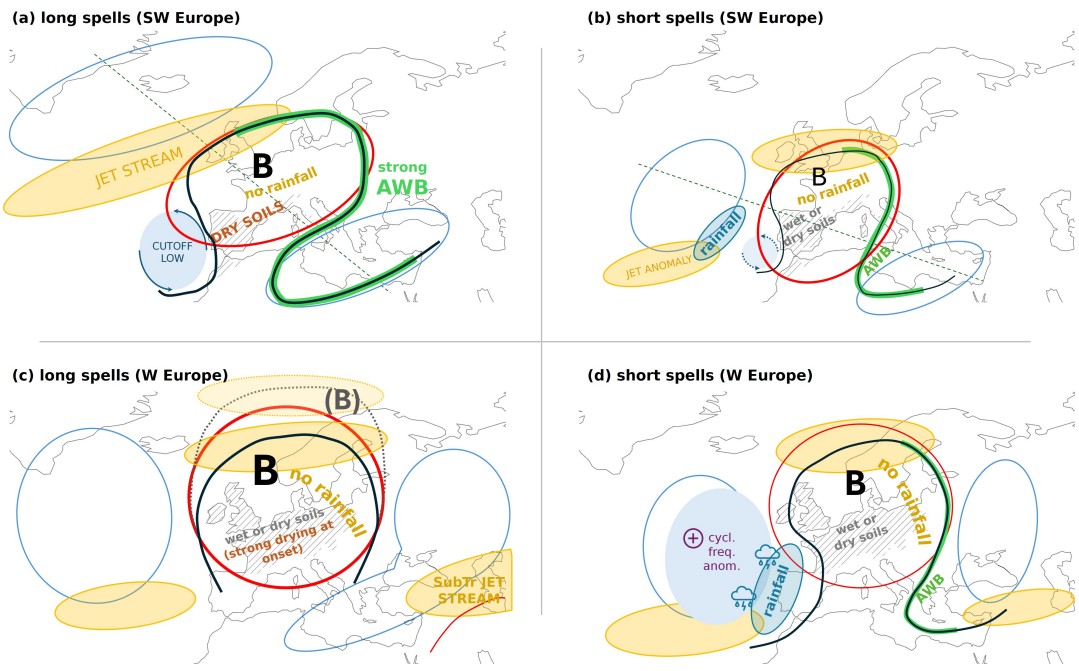

**Figure 7.** Summary schematic depicting idealised structure of long and short spells in SW Europe (a,b) and W Europe (c,d). It illustrates some of the common mechanisms presented and discussed in this study. The gray hatching marks the area affected by the extreme temperature event. The red and blue contours represent the positive and negative Z500 anomalies; the black line sketches the atmospheric block B; the yellow shapes denote the jet streaks; the green line represents anticyclonic wave breaking (AWB). The dotted black line and fainter yellow jet streak in panel c represent the potential larger meridional extent of the block (B).




## 3.4 Variability and typicality of hot spells

The composite analyses reveal characteristics of long and short hot spells. However, this method cannot capture the full diversity and complexity of flow structures and the varied configurations of driving mechanisms that lead to persistent hot spells. A closer look at individual long spells in both studied regions confirms this point (see Fig. 8). Each event is marked by its own SM conditions and atmospheric drivers. Each event has a distinct temperature evolution, with one or multiple local maxima. Some events resemble intense mega-heatwaves, while others consist of shorter waves in quick succession. Each spell is driven by its

own mixture of drivers, which can occur at various stages throughout the event, with different lifespans. This is in agreement withWehrli et al. (2019), who showed that the contributions of key physical drivers of heat extremes vary from one event to the next.

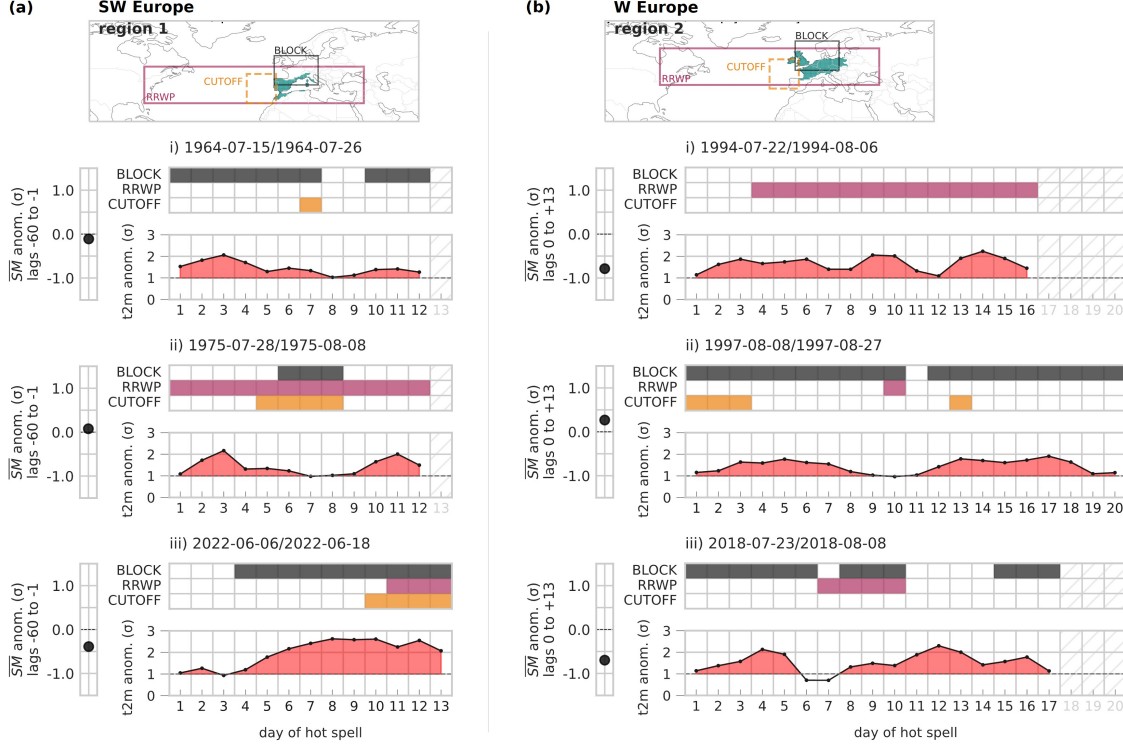

**Figure 8.** Three select persistent events for both SW Europe (left) and W Europe (right). Each case is made up of three subplots: the left sub-panel shows the average SM conditions in the 60 days before the event (region 1) and during the first 2 weeks after onset (region 2); the bottom sub-panel depicts the T2M evolution during the event in red shading; the top sub-panel illustrates the presence of three atmospheric drivers for each case (color filled boxes). The maps in the top row show the respective areas chosen to compute the proxies for the drivers. For a given day during the spell, a block (black) / cutoff (orange) is present when the feature covers at least 20% of the boxed area. RRWPs are accounted for when the region-averaged R metric anomaly exceeds its seasonal $80^{th}$ percentile.





In addition to the small sample size, our definition of the hot spells using a relatively low threshold of $1\sigma$ can be a source of variability. Using large deviations theory (Lucarini et al., 2023; Noyelle et al., 2024) show that as heatwave intensity in-
creases, the variance between the trajectories leading to the extreme decreases. In other words, typicality results from extreme magnitude: the higher the T2M anomaly, the greater the dynamical similarity.

Figures 9 and 10 provide an summary of each long and short event in SW Europe and W Europe, respectively, highlighting the relative presence of blocking, RRWPs, and cutoff lows in proportion to the event duration. Importantly, they show that the selected mechanisms tend to be present to some degree in the long spells. In SW Europe, blocking is detected during 81%
(37%) of long (short) spells, while in W Europe, blocks are present in 87% (59%) of long (short) spells. For RRWPs, these values are 50% (35%) for long (short) spells in SW Europe, and 60% (15%) for long (short) spells in W Europe. Although the composites showed that cutoffs are not so relevant for long spells in W Europe, these structures are present half of the time, and only in 23% of short spells. For long spells in SW Europe, these structures are present in all but one case and in around 40% of short spells. When interpreting and comparing these proportions, as well as the box sizes in the figures, it is important
to consider the differing sample size and duration of the two hot spell categories.

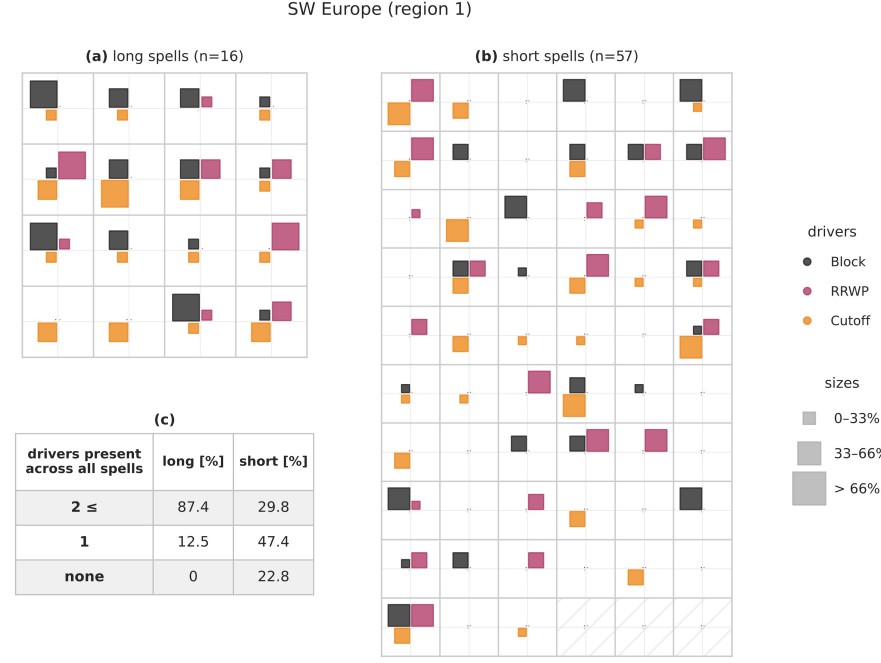

**Figure 9.** Visual representation of the relative frequency of blocks, cutoffs, and RRWPs during (a) long and (b) short hot spells in SW Europe. The size of each square corresponds to the prevalence of each driver in proportion to the event's duration. Table (c) illustrates the proportion of spells characterized by the presence of more than two, exactly one, or no drivers.

Generally, long spells seem to mostly be characterised by the presence of more types of drivers, compared to short spells (Figs. 9-10c). In both regions, at least 80% of long spells show a combinations of two or more drivers, compared to around



30% during short spells. Long spells in SW and W Europe, for instance, are often associated with simultaneous recurrence and quasi-stationarity in the large-scale flow. In other words, both transient wave activity in the form of RRWPs and near stationary

blocking co-occur to generate surface persistence.

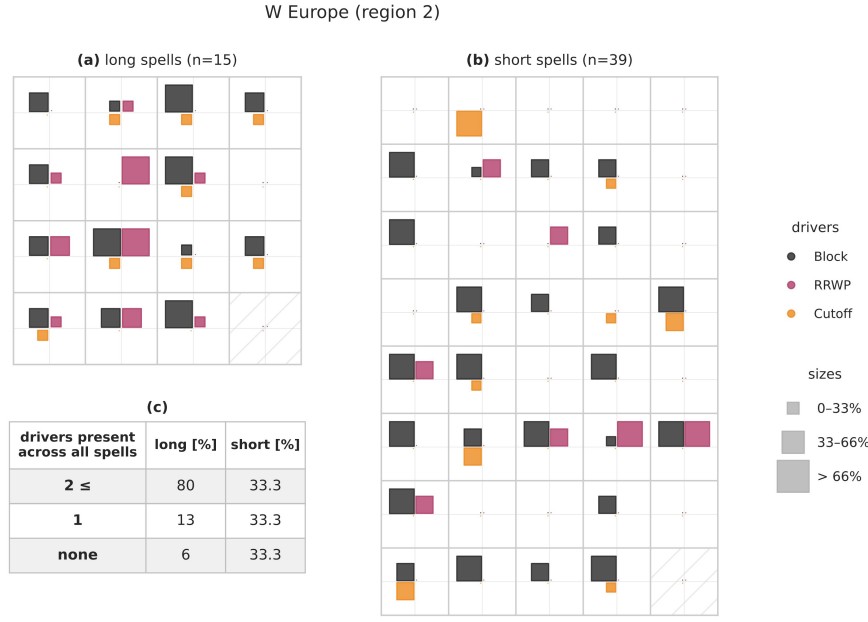

**Figure 10.** Visual representation of the relative frequency of blocks, cutoffs, and RRWPs during (a) long and (b) short hot spells in W Europe. The size of each square corresponds to the prevalence of each driver in proportion to the event's duration. Table (c) illustrates the proportion of spells characterized by the presence of more than two, exactly one, or no drivers

In summary, duration represents an important dimension of temperature extremes that adds a level of complexity in understanding the associated weather and climate dynamics, with implications for their predictability (see e.g., Lembo et al., 2024). Accurately forecasting such extreme events on S2S timescales remains a major challenge (White et al., 2022). A comparative framework such as the one used in this study offers a valuable approach to begin addressing some of the questions and hurdles.

## 4    Conclusion & Outlook


We investigate the differences between long (persistent, 12 to 26 days) and short (4 to 5 days) hot spells in SW and W Europe. We first introduce a regionalisation of summer extreme temperatures using data from the ERA5 reanalysis between 1959 and 2022. The regionalisation is based on the Jaccard distance as a measure of similarity for grouping locations over land into clusters that tend to experience 3-week T2M extremes simultaneously. We define long- and short-duration events for each

cluster and compare composites of several relevant variables and drivers between these two sets of events in the selected regions. Additionally, our analysis draws out factors common to all persistent hot spells.



Long spells in both SW and W Europe are on average associated with stronger and lower wavenumber upper-level Rossby wavetrains, compared to short spells. Lower WN Rossby waves are more stationary than higher wave-number and can thereby increase stationarity of surface weather systems. Long spells in SW Europe, specifically, are characterised by significant soil dryness up to two months prior and a significant cumulative SM deficit at the strat of the events, which delays drought recovery and prolongs positive land surface feedbacks. A pronounced meridional wave pattern, likely driven by strong anticyclonic RWB, deflects the jet stream northward, steering cyclonic activity further north. Frequent blocking and upstream cutoff lows are prominent features during long spells in SW Europe, both of which contribute to persistent hot surface weather conditions.

In contrast, long spells in W Europe have a different relation to SM, where strong land-atmosphere coupling and consequent soil desiccation during the event plays a critical role. The distinction between long and short spells in this region is further marked by the high frequency of large, long-lived blocks and frequent RRWPs during the long spells. The regional distinctions in the relative importance of dynamical features underscore the usefulness of our regionalisation. Short spells are characterised by a more zonal upstream flow structure, a stonger jet upstream of the hot spell region and a more active storm track upstream. All are indicators of a more transient flow over the eastern North Atlantic and the imminent arrival of disturbances that bring lower temperatures and potentially rain to the hot spell region and ultimately terminate the spells.

A look into the individual events also reveals that long spells are more often characterised by the co-occurrence of multiple drivers (blocking, RRWP, cutoffs), involving their complex interaction over subseasonal timescales. Ultimately, to fully characterise these persistent events and the intricate dynamics that shape them, additional data is necessary. Despite this main limitation, our methodology provides a useful comparative framework with which to parse the dynamical components of persistent hot spells and understanding the factors their long duration. Extending our analysis to large-ensemble climate model simulations could offer a significantly larger sample size, leading to even more substantive conclusions.

Finally, other mechanisms not examined in this study may have mattered for affecting the duration of hot spells. Drought self-propagation and remote influences of SM should be further explored in connection with hot spell longevity. Other climate drivers modulating at longer timescales include anomalous SSTs, tropical convection, and specific teleconnections that could force, even phase-lock, wave activity over Europe. Studying the vertical profile during these events would also provide a more seamless view of the link between persistent surface weather, boundary layer dynamics, and upper-level circulation.

*Code and data availability.* A repository of the python code related to this study is available at http://www.github.com/dpappert/persistent-hotspells. ERA5 reanalysis data can be downloaded from the Copernicus Climate Data Store (Hersbach et al., 2023).

*Author contributions.* DP processed the data, performed the analysis, drafted the manuscript and designed the figures. AT, DC, MV and OM provided critical feedback and helped shape the research, analysis, and manuscript. OM supervised the project.



*Competing interests.* The authors declare that they have no conflict of interest.

*Acknowledgements.* This research has been supported by the Schweizerischer Nationalfonds zur Förderung der Wissenschaftlichen Forschung (project PERSIST-EU, 207384). The work of MV was supported by the "COESION" project funded by the French National program LEFE (Les Enveloppes Fluides et l'Environnement). MV also benefited from state aid managed by the National Research Agency under France
2030 bearing the reference ANR-22-EXTR-0005 (TRACCS-PC4-EXTENDING project).



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
