# Peer review of "Long vs. Short: Understanding the dynamics of persistent summer hot spells in Europe"

_EGUsphere, 2024_

## Referee Comment (RC1)

**Review of „Long vs. Short: Understanding the dynamics of persistent summer hot spells in Europe" by Pappert et al. submitted to Weather and Climate Dynamics**

**Summary and general assessment**

Pappert et al. analyse the dynamics of long-lasting (> 12 days) and short-lived (4-5 days) summer hot spells in southwestern and western Europe. While hot spells are extensively studied in the existing literature, the authors focus here on the spatio-temporal characteristics of an ensemble of various dynamical processes in different European regions, with a special focus on the difference between short and long lasting hot spells. These processes include, amongst others, atmospheric blocks, recurrent Rossby wave packets (RRWP) and cut-offs. In addition to the dynamical, more short lived processes, they also incorporate the analysis of the longer-lasting climate variable soil moisture prior to the hot spells. The main results of the study are, amongst others, a stronger link between antecedent soil moisture and long-lasting hot spells in southernwestern Europe compared to western Europe. In the latter region, long-lasting hot spells are associated with a more stationary upper-level flow, together with a high blocking frequency and high amount of RRWPs. Short hot spells in western Europe feature a higher than average upstream cyclone frequency, which underlines the important role of cyclones for terminating heat waves in this area.

Overall, the study is very well-written, very nicely analysed and figures are of high quality. I have only a few minor comments which should be addressed prior to publication.

**Suggestion:** Accept with (very) minor revision

**Minor comments:**

1. L121: is there a reason for a 3-week, non-overlapping period? How does this influence your results, in particular the short heat waves?
2. L146: how do you define a heat wave day? Is there a minimum spatial threshold or is one grid point sufficient?
3. L161: for long-lasting heat spells: do you use the same 4-5 day period for all meteorological fields?
4. Figure 1: there appears to be a typo in the box next to I. ($l\_4^*$ and $l\_3^*$ must be switched)
5. L210: the reader may be interested in the other regions too (maybe add some results to the supplementary material if it adds an added value to the study)
6. L259: parenthesis missing in the quotation
7. L286: remove point before Rossby wave
8. L331: cutoffs > cutoff
9. L361: insert blank space before Wehrli et al.
10. Figure 9c/10c: $\geq 2$ instead of $2 \geq$
11. Figure 9/10: the size of the squares … —> does it mean that e.g. in 66% of the heat wave it is dominated by a feature (e.g. a block)
12. L393: remove typo before Rossby waves (WN)

---

## Author Response (AR1)

We thank referee #1 for their review and various suggestions to improve the manuscript. In the following we will respond to the comments listed by the referee, which are written in italics. Our responses are written in bold.

**Minor comments:**

*1. L121: is there a reason for a 3-week, non-overlapping period? How does this influence your results, in particular the short heat waves?*

**The choice of 3-week, non-overlapping periods has two motivations:**
**First, we choose 3-week periods to focus on the subseasonal timescale in which we require the co-occurrence of temperature extremes to be reflected in the resulting clusters. For the same level of event synchronicity (12.5%), clusters resulting from 3-week periods are larger and fewer than with 2- or 1-week periods. This choice is made somewhat redundant by the fact that region size and number can additionally be subjectively changed by altering other parameters such as the threshold for the dendrogram truncation or even for the binarisation before the clustering (see Section 1 of the Supplementary Materials). In the end, the choice for the parameters was tuned to end up with regions of size and location that best matched European heatwave research, whilst still being objective in its grouping of grid cells and demarcation of clusters based on observed extreme temperature variability. Long-lasting or short events can be defined for regions of most sizes. That said, one could speculate that region size would affect the distribution of event durations. In particular, very short heatwaves (1-2 days) probably affect an even smaller fraction of the regions. Second, we choose non-overlapping periods to minimise the statistical dependence between different periods, which would affect the statistical significance testing we perform in our analyses.**

*2. L146: how do you define a heat wave day? Is there a minimum spatial threshold or is one grid point sufficient?*

**We define a hot day as a day exceeding the +1 standard deviation threshold of the region-averaged standardised anomalies, which were previously deseasonalised and detrended. We mention this threshold at l.150. As the other reviewer pointed out, this is an important sentence that seems rather casually mentioned, so we will give it more prominence in the paragraph by mentioning it at the beginning.**

**The threshold is based on the standard deviation of the time series resulting from the spatial average of the standardised anomalies in the region. The exceedance was not defined based on a minimal spatial requirement. However, we did explore other methods to define a hot day and how these affect the distribution of hot spell durations, which are briefly discussed in Section 2 of the Supplementary Materials.**

3. *L161: for long-lasting heat spells: do you use the same 4-5 day period for all meteorological fields?*

**We use a different random seed for each meteorological field, meaning that we subsample different subperiods each time. Even with this approach, we observe a consistent correspondence between the composites across variables (i.e. in almost all cases. During the analysis, we also subsampled different periods for the same meteorological field, and this did not alter the qualitative patterns or the statistically significant areas.**
* * *
4. *Figure 1: there appears to be a typo in the box next to I. (I_4^\* and I_3^\* must be switched)*

**We understand how this may look like an oversight. In the logic of the schematic, the subspell period $I_3$ was selected prior to $I_4$. In other words, the sequence should not matter for the labelling as the subperiods are sampled randomly across all long spells. However, to avoid confusion, we will correct this, especially since the rest of the subspells appear to be ordered.**
* * *
5. *L210: the reader may be interested in the other regions too (maybe add some results to the supplementary material if it adds an added value to the study)*

**We have already included the same figures from Sections 3.2-3 of the main manuscript for the other regions in Section 4 of the Supplementary Materials, for those interested in reviewing them.**
* * *
6. *L259: parenthesis missing in the quotation*

**Added.**
* * *
7. *L286: remove point before Rossby wave*

**Removed.**
* * *
8. *L331: cutoffs > cutoff*

**Corrected.**

9. *L361: insert blank space before Wehrli et al.*

**Added.**

10. *Figure 9c/10c: ≥ 2 instead of 2 ≥*

**Corrected.**

11. *Figure 9/10: the size of the squares … —> does it mean that e.g. 66% of the heat wave it is dominated by a feature (e.g. a block)*

**Yes, that is the correct way to interpret the square size.**

12. *L393: remove typo before Rossby waves (WN)*

**'WN' is meant to come before 'Rossby wave', as it refers to the latter's wavenumber. This part of the sentence is meant to be read as: "Lower WN Rossby waves are more stationary than higher WN waves and can thereby enhance the persistence of surface weather systems." The sentence will be rewritten in this way for clarity. Also, we will make sure to be more consistent with the use of acronyms.**

We thank referee #2 for their thoughtful, engaging and constructive feedback. We appreciate the suggestions to improve the quality of the manuscript. In the following document, we hope to adequately respond to both the general comments and specific comments. Referee comments are written in italics, while our responses are in bold.
* * ** * *
**General comments:**

1. *The motivation of the paper is presented in a somewhat misleading fashion. The authors repeatedly highlight the impacts of heatwaves as motivation for their work, but then focus their analysis on moderate events (which they in fact term "hot spells" rather than heatwaves". I do find the analysis of the hot spells interesting, but I would suggest a more nuanced framing of the introduction. In particular, the framing should recognise that the events the authors analyse are not necessarily high-impact ones from a societal viewpoint. If impacts were the only consideration, then the authors should revise their definition of hot spells.*

**While the events we focus on in our study might not necessarily be high-impact in the same way as the newsworthy mega-heatwaves, they are still capable of putting stress on natural and societal systems by virtue of their longevity. They might surpass the hot day threshold by little or a lot, but it is their continued exceedance over many days/weeks that does not allow these systems respite and thus potentially making the cumulative impact quite high. This could negatively impact the recovery of a system or sector following an event (Flach et al. 2021, Polt et al. 2023). For some parts of the midlatitudes, the difficulty in acclimatising to a longer-lasting shift of above-normal temperatures will increase mortality (Anderson and Bell 2011). Polt et al. (2023) also argue that the duration of a heat event affects the societal response independent of, and additionally to, its intensity. Heatwaves in Germany, for instance, tend to show the strongest societal response between 2 weeks and 2 months.**

**We will make sure to better drive home this particular aspect of heat impacts in the introduction/motivation.**
* * *
2. *While I understand that the focus of this study is on persistence rather than intensity, it would still be interesting to provide some information on the intensity of the events that you analyse in the main text, e.g. in Sect. 3.1. This would enable some understanding of what a "hot spell" entails in practice, and enable to relate the events being analysed to the typical intensity of recent European heatwaves.*

**The relatively low threshold (1sd) chosen to define hot spells in this study means that the detected hot days range from uncommon to somewhat rare in terms of their magnitude. For the six clusters, this corresponds roughly to the 85th percentile of the region-averaged anomalies. Röthlisberger et al. (2019) used this threshold, applied on**

a gridpoint scale, referring to such events as 'moderate hot spells'. To provide additional context on the magnitude of these events, we have included Figure R1.

[Figure]

**Figure R1: Distributions of mean long v. short hot spell intensity (a) and the intensity of individual hot days within those spells (b) across all six clusters from the regionalisation.**

Figure R1 shows a range of intensities across events, with most corresponding to the region-averaged 90th percentile – a threshold that has often been selected to define hot days for heatwaves (Domeisen et al. 2023). The intensity distributions also reveal an interesting regional difference: visually, there is no noticeable difference in the median intensity of long versus short spells in regions 2 and 3. In contrast, for regions 4 and 5, the shift toward higher anomalies for long hot spells is pronounced.

Furthermore, as mentioned in the manuscript, our definition of hot spells allows for identified waves of heat to be merged into one event if occurring very close in time. This step introduces non-hot days that can lower the spell average, as evidenced by the presence of individual days that classify as below the 80th percentile.

We will ensure that some of this information is included in Table 1 of Section 3.1, by adding columns with mean spell values and confidence intervals. Additionally, we will add the figure above to the Supplementary Materials document and refer readers to it in the main manuscript for further details regarding the intensity of these events. This should help contextualise what a 'hot spell' entails in practice, and assist in relating these events to the typical intensity of recent European heatwaves.

3. *Throughout the paper, the authors equate persistence or stationarity of the large-scale flow with duration of the hot spells. However, this relationship is not as straightforward as the authors implicitly make it. For example, Holmberg et al. (2023) argued for a nuanced and variable relationship between the persistence of the large-scale flow and European temperature extremes.*

**Thank you for this comment. You raise an important point, and we appreciate the opportunity to clarify this aspect of our analysis. It is not our intention to suggest a simple 1:1 relationship between persistence of the large-scale atmospheric circulation and the persistence of surface temperature extremes. We would also like to note that, regardless of our somewhat diverging characterisation of persistence, Holmberg et al. (2023) makes the link between persistent atmospheric configurations and warm-temperature extremes, without focusing on the latter's persistence beyond the synoptic timescale, which *is* the subject of our study. That said, we fully agree with the need for a nuanced understanding of this relationship.**

**Our main argument drawn from the composite analysis of atmospheric fields in Section 3.3 is that, on average during the lifetime of long hot spells, there are certain structural configurations of the upper-level large-scale dynamics that favour this prolonged surface weather. Through the literature, we know that such configurations have indeed a strong link to flow stationarity.**

**Following this, Section 3.4 provides necessary nuance and elaboration. We demonstrate that this link is not as straightforward as suggested by the composites. Specifically, we show that long spells typically exhibit flow quasi-stationary and/or recurrent flow patterns, both persistent circulations capable of causing persistent surface weather. Indeed, around half of long spells feature synoptic-scale transient recurrent Rossby wave packets (RRWPs), sometimes accompanied by blocking, sometimes not. This underscores that the flow cannot simply be described as 'just stationary' in all cases.**

**Moreover, we emphasise that these persistent circulation features are often present only for portions or moments of the hot spell's lifetime, often overlapping or occurring in various sequences. This suggests that the flow associated with long spells does not need to remain wholly unchanged throughout its entire duration to contribute to its overall persistence; rather, the intermittent (or combined) influence of the drivers across different periods of the spell can still sustain prolonged surface weather.**

**We will ensure these points are better highlighted throughout the manuscript and that our statements convey the necessary nuance while remaining clear and unambiguous.**

**Specific comments:**

*Note that the list of typos and poorly phrased passages below is not comprehensive. I would recommend a thorough proof-reading of the text before a revised version is resubmitted.*

**Many thanks for pointing out the typos and oversights, which shall be corrected. The whole manuscript will be proof-read and more carefully examined for such mistakes.**
* * *
*l.6 This is poorly phrased. When you write "Temperatures are averaged across these regions for an analysis of hot spells in SW and W Europe.", it sounds as though you use temperature information from all 6 regions for the analysis of 2 of them.*

**Agreed it is unclear. We shall rephrase the sentence to avoid this misunderstanding.**
* * *
l.13 - *It is unclear why a zonal flow must necessarily mean a more transient flow. In fact, there are studies that argue that the zonal large-scale flow is the "default" state of the atmospheric circulation, with blocking representing an unstable fixed point (e.g. Faranda et al., 2016).*

**We realise from the reviewer comment that the expression transient is understood slightly differently in different sub-communities. We were using the term transient coming from a Rossby wave dynamics point of view, meaning: transient vs. stationary waves, the reviewer interpreted the sentence more from a system dynamics point of view. We will clarify "our" meaning by reformulating this sentence slightly.**
* * *
*l.29 Do the authors mean excess deaths?*

**Corrected.**
* * *
*l.33 The citation of Tuel and Martius (2024) is misleading. The paper does not analyse impacts, and does not provide any evidence of a link between duration of hot spells and severity of impacts, beyond citing other literature on the topic.*

**We shall provide better references that establish a link between the duration of hot spells and the severity of their impacts.**
* * *
*l.49 "that" should be "the"*

**Corrected.**
* * *
*ll.64 – 65 This is an incomplete sentence.*

**The line mentioning RRWP belongs to the previous sentence on weather recurrence. We will connect the two sentences.**
* * *
l.100 - *Why start from 1959? This is neither the start of the available ERA5 data nor the start of satellite assimilations. Also, since the authors struggle with sample size, adding the summers of 2023 and 2024 could provide a few extra persistent hot spells.*

**It is true that adding the currently available additional years would provide a handful of additional long spell cases for our analysis. However, at the time the data for this study was gathered – with the many variables post-processed and feature detection algorithms applied – it was early summer 2023, and the ERA5 dataset then only offered a back extension to 1959. Given the substantial time already invested in downloading, processing, and analysing the data, it was decided to proceed with the 64 years of data available at that point, rather than later revisiting the entire processing pipeline to include more years.**
* * *
l.114 - *This certainly works well for temperature and other smooth variables, but given that only 8 years are used, is this enough to give a smooth climatology also for noisy variables such as TP?*

**Thank you for raising a valid point. Following your comment, we computed the total precipitation (TP) composites based on anomalies that were calculated centered on a 20- and 30- year moving window. Figure R2 shows that using a 20- or 30-year running window does not affect the pattern and structures found when using an 8-year running window. Hence, for consistency, we use the 8-year running window for all variables.**

[Figure]

**Figure R2: Total precipitation composites for long (left) and short (right) spells, computed using anomalies derived from different climatological windows: 8-year (a,b), 20-year (c,d), and 30-year (e,f).**
* * *
l.121 - *Why do the authors choose non-overlapping periods?*

**Depending on the autocorrelation of the temperature series, overlapping periods could introduce temporal dependencies that would complicate the analysis and make it difficult to distinguish different events. To avoid such statistical issues, we chose non-overlapping periods. The referee raises a valuable point, which presents a challenge that could be addressed in future work.**
* * *
*l.123 95th percentile of what? Of the distribution of 3-week averaged temperatures? Please clarify in the text.*

**Will clarify.**
* * *
*l.138 Verify that the figures are referred to consecutively in the text. Fig. 2 appears to be referred to before Fig. 1.*

**This will be adjusted accordingly.**
* * *
*l.145 "and" should be "to"*

**Corrected.**
* * *
l.150 - *Why do the authors select a threshold in units of standard deviations here, but used a percentile threshold for the regionalisation? Also, the 1 sd threshold comes out of nowhere, and some context for it should be provided; for example, a sentence stating explicitly that a 1 sd threshold is used to define the hot spells.*

**The two computations (regionalisation and hot spell definition) address two different questions. We performed the regionalisation using percentiles, as they rely solely on the rank of values, and as such allow us to remain more generic/flexible in the face of potential distributional differences across the gridcells of the entire European domain.**

**For the hot spell thresholds within the now defined regions, we compared several options, both in percentiles and standard deviation units. Given the amount of data we proceeded the study with, the +1$\sigma$ threshold satisfied our need for sufficient persistent events, while still retaining sufficiently moderate hot days. Such a threshold has previously been used to study heatwaves, for instance in Weirich-Benet et al. (2023), where weekly averages exceeding this threshold are regarded as "high temperature anomalies".**

**We will add a sentence at the beginning of this paragraph specifying the threshold used to define hot days/spells.**
* * *
l.158 *The reference to Tuel and Martius (2023) is again potentially misleading here. The authors again use a paper that does not conduct any explicit analysis of impacts to support a sweeping statement on impacts of hot spells. It would be more instructive for the reader if the authors cited some of the papers that Tuel and Martius (2023) use to support the statements that they make on impacts.*

**We shall cite studies that more explicitly tie long-lasting heat extremes to their impacts, such as those referenced in our reply to General Comment #1.**
* * *
l.185 - *Here the authors state that they "consider grid cells significant if they are identified as such in at least two-thirds of the 100 subsampled composites". However, in some of the later figures they use two different types of stippling to highlight both the 1/3 and 2/3 threshold cases.*

**Thank you for pointing out this oversight. We will modify l.185 to include the explanation of the 1/3 stippling that is illustrated in Figs. 5-6.**
* * *
l.187 Here the authors could add a clarification that the stated confidence levels for the long spells should not be interpreted at face value, since the authors show cases where a given fraction of a sample satisfies a given confidence level. In other words, the stippling in the later figures for the long spells cannot be interpreted statistically in the same way as a 95% confidence level normally would be.

**We will insert a statement on this matter to avoid misinterpretations. The stipplings are to be interpreted differently, though in both cases, their objective is to point out regions where there is a signal.**
* * *
l.203 - *This makes the choice of regions that the authors focus on in the main text debatable. Why pick the two regions with the fewest long spells as focus regions if the authors repeatedly highlight sample size as a challenge for their analysis?*

**The choice of regions in the main text was not based on the availability of long spell events, which is a limitation for all six regions. We selected SW and W Europe because they are geographically close, yet at two distinct latitudinal bands, each with different blocking frequencies and soil moisture regimes, making them suitable for an interesting comparison. While these regions have fewer long spells, they still provide meaningful insights into the broader analysis. Considering a few additional long spells would make the composites somewhat more representative, but it is unlikely to significantly impact the robustness of the analysis or the underlying arguments. Figures for the other four regions are available in the Supplementary Materials (Sect. 4), and these regions can be explored in future work.**
* * *
*Fig. 3 I find the "Q" labels along the y-axis in panel (b) misleading. I would assume that these refer to "quantiles" or "quartiles", while according to the caption they instead signify deciles.*

**The caption refers to 30th, 20th, and 10th percentiles, which are the equivalent to the Q0.30, Q0.20, and Q0.10 on the x-axis of Figure 3b. We will refer to them as quantiles in the caption to be consistent.**
* * *
l.240 What do the authors mean by "contrasting spread"?

**We mean to say that, for region 2, the thinner boxplots representing spell onset contrast in the extent of their spread. Soil moisture values at the beginning of long**

**spells are very unlikely to be above climatology, whereas during short spells they can take on a broader range of values.**
* * *
*I.241 "Bulk of values". Do the authors mean bulk of events?*

**Corrected.**
* * *
*I.245 What does "significantly extreme" mean? The figure shows values significantly different from zero.*

**Corrected.**
* * *
*II.262/4 - This is an interesting consideration. Does this suggest that long spells are long because they are weak while short spells may be short by virtue of their intensity, as the very high temperatures are a potential trigger for convection?*

**This sentence is somewhat speculative, and we do not suggest that short spells are short because they are more intense – at least not more so than long spells (depending on the region in question it is actually the other way around, see Fig. R1). Our understanding of the link between convection and the end of a heatwave, based on the study by Zhang and Boos (2023), is that temperatures can only rise so high before the atmosphere becomes unstable and triggers convection, even in a predominantly dry atmosphere. When temperatures during a hot spell reach this intensity, convection could occur and potentially end the hot spell sooner than if temperatures had remained more moderate. This could theoretically happen after any number of days. To avoid misinterpretation or confusion, we suggest removing this sentence.**
* * *
*I.281 - The authors should support their statement about persistence of the flow. Several of them have previously worked on this topic, and are certainly aware of appropriate references that could be used here.*

**Again, we look at things from a Rossby wave perspective and the term "flow" includes most prominently also Rossby waves. From the system dynamics point of view this statement might be less obvious. We will certainly add relevant references, such as Hoskins and Ambrizzi (1993) and Hoskins and Woolings (2015), to support our statement.**
* * *
l.285 "there is has"

**Noted. Should be "there is".**
* * *
l.286 Remove the extra full stop

**Removed.**
* * *
l.298 - *Not everyone would agree with this statement, and in fact Lucarini and Gritsun (2020), identified blocking as not necessarily being a persistent configuration; see also my comment on l. 13. Holmberg et al. (2023) also argued against the notion that blocked large-scale atmospheric flows must be a priori anomalously persistent.*

**What we meant with blocks being synonymous with persistence is that they are typically equated with large-scale anticyclonic conditions that persist beyond their usual duration. In other words, blocks consist of persistent geopotential height anomalies. A 'persistent blocking event', which we are *not* referring to, would suggest a long-lived block. Accordingly, the term 'persistence' should be interpreted with care, as its meaning can vary depending on the characteristic timescale of the feature being described. We shall add a clarifying statement in this regard in the text.**

**Furthermore, we would like to note that the persistence metric used in Holmberg et al. (2023) finds a significant link between summertime heatwaves and mid-tropospheric persistence, consistent with the presence of atmospheric blocks. It is only when using sea-level pressure (SLP) as a proxy for atmospheric flow patterns that they do not find this persistence link with extreme temperatures, which makes sense given the reduced relevance of SLP in summer compared to winter because of local heat lows.**

**We will likely include a sentence in the introduction to clarify that when we refer to persistence, we are talking about episodic persistence and weather system persistence.**
* * *
l. 304 - *Where can the reader find information on blocking size in the figure?*

**Admittedly, the blocking frequency composites alone do not prove that individual blocking systems during the long events are larger. However, it shows that the overall positive blocking frequency signal extends over a wider area during persistent spells.**

**Figure R3 would indeed be a better illustration of this point. When comparing the mean area covered by blocks during the lifetime of individual spells, we observe that a greater portion of long spells than short spells are accompanied by larger blocks, as**

evidenced by the difference in the median. A more precise phrasing would be that long hot spells are less likely to have no/small blocks. In contrast, while it is less common for short hot spells to be associated with larger blocks, it is certainly still possible. The statement comes down to a difference in the median, with the extent of this difference depending on the blocking index being used.

[Figure]

Figure R3: Boxplots showing the average fraction of area covered by blocking during the lifetime of long and short spells for SW Europe (region 1, top row) and W Europe (region 2, bottom row). The leftmost panels display results for blocking detected using vertically averaged integrated potential vorticity (VIPV), while the rightmost panels show results from the blocking index based on geopotential height anomalies at 500 hPa (Z500).

Based on this one could conclude that block size, and by implication quasi-stationarity, can contribute to the persistence of near-surface hot spells. However, as in the case of soil moisture anomalies, this aspect alone does not solely account for the difference in long and short hot spells and is subject to other conditions/configurations taking place.

We will include this figure in the manuscript in Section 3.3.2 to better support our argument.
* * *
*l.314 "VIPV": define the acronym.*

This should have been defined in l.106; here, we will change PV to VIPV.
* * *
*l.324 The statistically significant positive precipitation anomalies are also seen for the long spells, but the way the text is formulated presents this as a factor differentiating short hot spells from longer ones.*

The long spell composites for both SW and W Europe (Figs. 4,5) do *not* show statistically significant precipitation anomalies upstream of the hot spell regions, unlike the short spells. In fact, the current formulation should suggest this as a distinguishing factor.

*l.339 – 340 Why do the authors use a separate paragraph for these two lines?'*

**The separation is not intentional. The two lines will be part of the previous paragraph.**
* * *
*l.343 The sentence is missing an "a"*

**Added.**
* * *
l.347 - *Both the text and the figure give prominence to RWB, but this is something that the authors never explicitly diagnose. I would recommend: clarifying that the RWB is assessed visually, or computing RWB with some existing algorithm, or toning down RWB in the summary text and figure presented here.*

**Overturning RWB composites were computed as part of our analysis, detecting them using an algorithm from the same repository that was employed to track cutoffs. Yet, we ended up including these plots only in the Supplementary Materials (Sect. 3, Figs. S11-12). The presence of marked RWB is mainly relevant in the context of long spells over SW Europe (region 1). This can be assessed visually in the slanted pattern of the Z500 anomalies, suggesting anticyclonic RWB (see Fig. 11 in Tamarin-Brodsky and Harnik, 2024). The frequent appearance of cutoffs upstream of the hot spell region is also indicative of RWB activity. Both of these observations are corroborated by the frequency anomalies of overturning anticyclonic RWB in the composites, which are not included in the main text but are detailed in the Supplementary Materials.**

**In response to the comment, we will clarify in the manuscript that RWB is assessed visually and/or inferred in these instances and refer to the supplementary figures for confirmation. This adjustment should alleviate any concerns about overemphasising RWB and better justify its role in the argument.**
* * *
*l.361 There is a missing space*

**Added.**
* * *
*l.361 There is a missing space*

**Added.**

*Sect. 3.8 makes a number of interesting points and I enjoyed reading it.*

**Thank you!**
* * *
*l.367 "an summary"*

**Corrected.**
* * *
*l.393 Define the acronym "WN"*

**Changed to "wavenumber (WN)".**
* * *
*l.395 "strat"*

**"Start"**
* * *
*l.410 "the factors their long duration"*

**"the factors driving their long duration"**
* * *
*l.411 An alternative to using climate models could be to leverage reforecasts, which may better reproduce the synoptic and large-scale features discussed by the authors.*

**Indeed, that is a valid point. We will incorporate this suggestion, thank you.**
* * ** * *
**LITERATURE:**

- **Anderson, G. B. and Bell, M. L.: Heat Waves in the United States: Mortality Risk during Heat Waves and Effect Modification by Heat Wave Characteristics in 43 U.S. Communities, Environ Health Perspect, 119, 210–218, https://doi.org/10.1289/ehp.1002313, 2011.**

- Domeisen, D. I. V., Eltahir, E. A. B., Fischer, E. M., Knutti, R., Perkins-Kirkpatrick, S. E., Schär, C., Seneviratne, S. I., Weisheimer, A., and Wernli, H.: Prediction and projection of heatwaves, Nat Rev Earth Environ, 4, 36–50, https://doi.org/10.1038/s43017-022-00371-z, 2023.

- Flach, M., Brenning, A., Gans, F., Reichstein, M., Sippel, S., and Mahecha, M. D.: Vegetation modulates the impact of climate extremes on gross primary production, Biogeosciences, 18, 39–53, https://doi.org/10.5194/bg-18-39-2021, 2021.

- Holmberg, E., Messori, G., Caballero, R., and Faranda, D.: The link between European warm-temperature extremes and atmospheric persistence, Earth System Dynamics, 14, 737–765, https://doi.org/10.5194/esd-14-737-2023, 2023.

- Hoskins, B. J. and Ambrizzi, T.: Rossby Wave Propagation on a Realistic Longitudinally Varying Flow, Journal of the Atmospheric Sciences, 50, 1661–1671, https://doi.org/10.1175/1520-0469(1993)050<1661:RWPOAR>2.0.CO;2, 1993.

- Hoskins, B. and Woollings, T.: Persistent Extratropical Regimes and Climate Extremes, Curr Clim Change Rep, 1, 115–124, https://doi.org/10.1007/s40641-015-0020-8, 2015.

- Polt, K. D., Ward, P. J., Ruiter, M. de, Bogdanovich, E., Reichstein, M., Frank, D., and Orth, R.: Quantifying impact-relevant heatwave durations, Environ. Res. Lett., 18, 104005, https://doi.org/10.1088/1748-9326/acf05e, 2023.

- Röthlisberger, M., Frossard, L., Bosart, L. F., Keyser, D., and Martius, O.: Recurrent Synoptic-Scale Rossby Wave Patterns and Their Effect on the Persistence of Cold and Hot Spells, Journal of Climate, 32, 3207–3226, https://doi.org/10.1175/JCLI-D-18-0664.1, 2019.

- Tamarin-Brodsky, T. and Harnik, N.: The relation between Rossby wave-breaking events and low-level weather systems, Weather and Climate Dynamics, 5, 87–108, https://doi.org/10.5194/wcd-5-87-2024, 2024.

- Weirich-Benet, E., Pyrina, M., Jiménez-Esteve, B., Fraenkel, E., Cohen, J., and Domeisen, D. I. V.: Subseasonal Prediction of Central European Summer Heatwaves with Linear and Random Forest Machine Learning Models, Artificial Intelligence for the Earth Systems, 2, https://doi.org/10.1175/AIES-D-22-0038.1, 2023.

- Zhang, Y. and Boos, W. R.: An upper bound for extreme temperatures over midlatitude land, Proceedings of the National Academy of Sciences, 120, e2215278120, https://doi.org/10.1073/pnas.2215278120, 2023.

---

## Author Response (AR2)

We thank referee #2 for the additional suggestions to improve the interpretability of the manuscript. Referee comments are written in italics, while our responses are in bold. Lines in brackets refer to latest tracked-changes version.
* * ** * *
*1. I appreciate the authors' clarifications in the introduction. I would recommend a further edit on l. 51 when the authors refer to "exceptionally warm temperatures". As the authors specify in their reply, events of a similar intensity to those that they detect have previously been referred to as "moderate hot spells". They later refer in their replies to "sufficiently moderate hot days". "Exceptional" and "moderate" have very different connotations, and in the case of many of the events that the authors select, it would be more appropriate to refer to them as moderate rather than as exceptional.*

**(l. 44) Removed "exceptional" - has been changed to "sustained warm temperatures".**
* * *
*2. ll. 409-411 I appreciate the addition here, but this means that the sentence on ll. 410-411 now refers to the new statement "A driver need not necessarily be present …", while I assume that the authors would want it to refer to the sentence on ll. 408-409. It would be appropriate to also specify here that, given this variability in drivers and in their links to surface hot events, the persistence of a driver does not a priori guarantee the persistence of the associated surface hot event.*

**(ll. 398-9) We thank the reviewer for pointing out that inconsistency. The sentence at ll. 410-411 was swapped up a place, now appearing before the new statement "A driver need not necessarily be present …". Regarding the reviewer's latter comment, we appreciate the suggestion, but we are hesitant to make this claim explicitly, as our analysis does not provide direct evidence of cases where a driver persists while the associated hot spell does not. Nonetheless, we agree that this is a valuable consideration and warrants further investigation in future work.**
* * *
*3. Analysis time period: I understand the issue faced by the authors. In this case I would recommend being open about the reason for the choice of time period in the text.*

**(ll. 105-106) We have inserted a brief sentence in this regard.**
* * *
*4. Comment regarding non-overlapping periods. As for the previous comment, in the interest of interpretability of the methodological approach, it would be helpful to add a sentence to the paper explaining this.*

**(ll. 130-1) We have inserted a sentence explaining the choice of using non-overlapping periods.**

*5. I do not follow the explanation regarding why the authors select a threshold in units of standard deviations for the hot spells, but used a percentile threshold for the regionalisation. A standard deviation threshold implicitly adapts to different distributional forms and will correspond to different percentiles for different distributions. Why is this a problem for the regionalisation but not for the selection of hot spells?*

**We thank the reviewer for raising this apparent inconsistency. We would like to amend our original reply on this matter by stating that the regionalisation was performed in a first step, for which we used a threshold in percentiles. Subsequently, when it came to detect events, after trying multiple thresholds of both kinds (percentiles and standard deviation), we settled on one that would somewhat increase the number of cases within the reanalysis period that was acquired. This is the main reason, which was not motivated by other considerations/assumptions regarding distributions.**
**As the reviewer correctly points out, a threshold in standard deviation results in different percentiles for different distributions - this might ensure comparability across regions in terms of departure from local variability. In contrast, a threshold in percentiles would ensure comparability in terms of event rarity, but not necessarily in terms of how anomalous the values are relative to local variability - i.e. the 85th percentile would lead to detecting 15% of observed values at every location. That said, given that the +1σ threshold for the different regions happens to correspond approximately to the 85th percentile, the practical difference between the two approaches is minimal. Such considerations, however, did not guide the chosen methodology.**
* * *
*6. Related to the above: I again emphasise the importance of the methods not only being reproducible but also interpretable, and I would recommend that the authors add a sentence to explain their choice in the main text.*

**(ll. 161-2) The sentence now explicitly mentions the main reason for opting for +1σ as a threshold. As such, we have transparently explained the method and choices for reproducibility. Other researchers can recreate our results or choose different thresholds if they wish to.**
* * *
*7. Original comment on l. 158. The authors are again citing papers that do not analyse impacts, but focus exclusively on hazards. The sentence that they are supporting with these citations starts with: "From an impacts perspective", and touches on both weather persistence and impacts. One or two studies looking explicitly at impacts are required to support this – in addition to the already cited studies on persistent events.*

**(l. 170) We now also cite the following papers: Quandt et al. (2017), using heat indices, show that the persistent weather conditions during the 2010 Russian heatwave were indeed impactful. Though not with regard to heatwaves, Barton et al. (2016) show that the serial clustering (flow recurrence) of extratropical cyclones producing extreme precipitation have large impacts on flooding.**

*8. L. 210 I would state explicitly that the stippling should not be interpreted statistically as corresponding to a specified significance level. The current formulation referring to "the same way" does not explain the core point here.*

**(ll. 196-202) The sentence has been amended for improved clarification.**
* * *
*9. Choice of regions: the authors mention "a few additional long spells" in their replies yet write "average nearly twice as many long spells" in the paper. I would not equate "nearly twice as many" to as "a few", and would ask the authors to reconsider the reply they have provided to my earlier comment.*

**We appreciate the reviewer's interest in the results for other clusters and agree that interesting insights can be gained by including other areas. However, including results for more regions would make this already very long paper even longer. We therefore needed to choose a selection of results to present in the paper. We chose two cluster areas that show interesting differences based on the different latitudinal positions of the clusters and the different soil moisture regimes. Part of our choice is also governed by the fact that we live in some of these areas.**
* * *
*10. ll. 353 and following. I appreciate this addition, but am unclear as to why the size of the blocking system should reflect its quasi-stationarity (ll. 358). I would recommend a clarification in this respect.*

**We thank the reviewer for pointing this out. We have clarified the physical link between the size of the blocking system and its quasi-stationarity by elaborating on the dispersion relation for barotropic Rossby plane-wave perturbations at the beginning of said paragraph (ll. 338-340). Larger-scale systems correspond to lower wavenumbers, which are associated with slower intrinsic phase speeds. In regions of weak mean flow, such systems can become quasi-stationary (Hoskins and Ambrizzi 1993).**